# Spatiotemporal trends and coordination of agricultural carbon efficiency in the Yangtze River Economic Belt and Yellow River Basin, China: An analysis of influencing factors and green finance integration

**Jingjie Li** [ORCID]*, **Chenying Cui**

School of Science, Tianjin University of Commerce, Tianjin, China

* lxyljj@tjcu.edu.cn

**Editor:** Hesham M. H. Zakaly, Ural Federal University named after the first President of Russia B N Yeltsin Institute of Physics and Technology: Ural'skij federal'nyj universitet imeni pervogo Prezidenta Rossii B N El'cina Fiziko-tehnologiceskij institut, RUSSIAN FEDERATION

## Abstract

As China's second-largest source of greenhouse gas emissions, agriculture is essential to achieving the goal of "carbon peak" and "carbon neutrality." Based on the measurement of agricultural carbon emissions (ACE) and agricultural carbon intensity (ACI) in 19 regions along the Yangtze River Economic Belt (YEB) and Yellow River Basin (YRB) in China from 2001 to 2020, this paper first uses the super-efficiency SBM model to measure ACE efficiency from static and dynamic perspectives. Then, the coupling coordination degree (CCD) between ACE efficiency and green finance in each region of the two basins is explored. Finally, Grey Relation Analysis (GRA) is used to obtain the influencing factors of CCD. The following conclusions are drawn: (1) The ACE in the YEB is almost twice that of the YRB. The ACE of the two basins generally experienced a trend of first growth and then declined, but the peak time was different. The ACI of the two basins showed a trend of continuous decline, and the decline rate of the YRB was faster. (2) The ACE efficiency of the two basins showed an overall upward trend, and the growth degree of different regions was vastly different. From the factor decomposition, the technological progress (TP) of the two basins significantly impacts the total factor productivity (TFP). (3) The CCD of ACE efficiency and green finance in the two basins increased from near imbalance to barely coordination level, and the CCD of the YEB increased slightly faster. The CCD of the two basins has a spatial difference of "downstream > midstream > upstream." (4) Among the influencing factors of the CCD of the two systems, the influencing degree of the factors is as follows from large to small: quality of human capital, level of economic development, government regulation, scientific and technological innovation ability.

## 1. Introduction

Carbon emissions will cause global warming, produce a greenhouse effect, and seriously harm the human living environment. Controlling carbon emissions has become the consensus of the

**Data Availability Statement:** All relevant data are within the manuscript and its Supporting Information files.

**Funding:** This work is supported by The Tianjin Philosophy and Social Science Planning Program, China (Grant numbers TJTJ23-001).

**Competing interests:** The authors have declared that no competing interests exist.

international community [1]. According to the Second Biennial Update Report on Climate Change of the People's Republic of China, agricultural activities are China's second-largest source of greenhouse gas emissions [2]. At the same time, the YEB and the YRB flow through a total of 19 provinces, with an area of 641 million acres of cultivated land, accounting for 32.08% of the country's total cultivated area, accounting for about 80% of the country's gross agricultural product, which is crucial to the development of China's agriculture. However, with the rapid economic and social development of the YEB and the YRB, the ecological environment has come under tremendous pressure. The ecological barrier protection needs to be improved, and the ecological protection mechanism needs to be better, resulting in a large ACE and the unbalanced development of the basin. Effectively controlling the growth of ACE in the two basins can restrain the growth of total carbon emissions [3]. There is a huge funding gap to achieve carbon emission reduction, so how do we coordinate the realization of carbon emission reduction targets? China is the only economy in the world that constructs a top-down green financial system, and green finance is an essential path to achieving carbon emission reduction. It is a practical issue of great significance to give full play to the role of green finance in ACE reduction and promote green finance to incline to the agricultural field. In order to better practice the concept of green development, cope with climate change, and improve the efficiency of resource utilization, the Central Committee of the Communist Party of China and The State Council issued the Opinions on Fully, Accurately and Comprehensively Implementing the New Development Concept to Achieve Carbon Peak and Carbon Neutrality in 2021, requiring the active development of green finance and the establishment of a sound green finance standard system [4]. Improving ACE efficiency is critical to achieving a good interaction between ACE reduction and economic development. Facing the dual strategic goals of improving ACE efficiency and developing green finance, in-depth research on ACE and ACE efficiency vigorously promotes the synergistic relationship between carbon emission efficiency and green finance and finds out the influence mechanism of each factor on the CCD between the two systems. Those have important theoretical significance and practical value for realizing economic development and ecological protection.

Therefore, this paper will study ACE in the YEB and the YRB from new perspectives, aiming to solve the following questions: (1) What are the levels and trends of ACE and intensity in the YEB and the YRB? (2) What is the ACE efficiency of the two basins from static and dynamic perspectives? (3) What is the CCD between ACE efficiency and green finance in the two basins? (4) What is the action mechanism of the factors affecting the CCD in problem (3)? The main content framework of this paper is shown in Fig 1.

The rest of this article is arranged as follows. Section 2 is a review of the literature. Section 3 describes the methods and models. Section 4 gives the source of research data and the selection of indicators. Section 5 gives the empirical analysis. Finally, Section 6 summarizes the research, gives relevant suggestions, and puts forward a discussion.

In order to facilitate readers' reading, this paper organizes the abbreviations of nouns into a table, the specific table is shown in Table 1.

## 2. Literature review

To emphasize this paper's research value and relevance, we review relevant studies on the Measurement of ACE in Section 2.1 and then introduce the methods of ACE efficiency in Section 2.2. Meanwhile, CCD analysis and Influencing factors analysis are summarized in Section 2.3 and Section 2.4. Finally, Section 2.5 gives a summary and discuss the contribution of the paper.

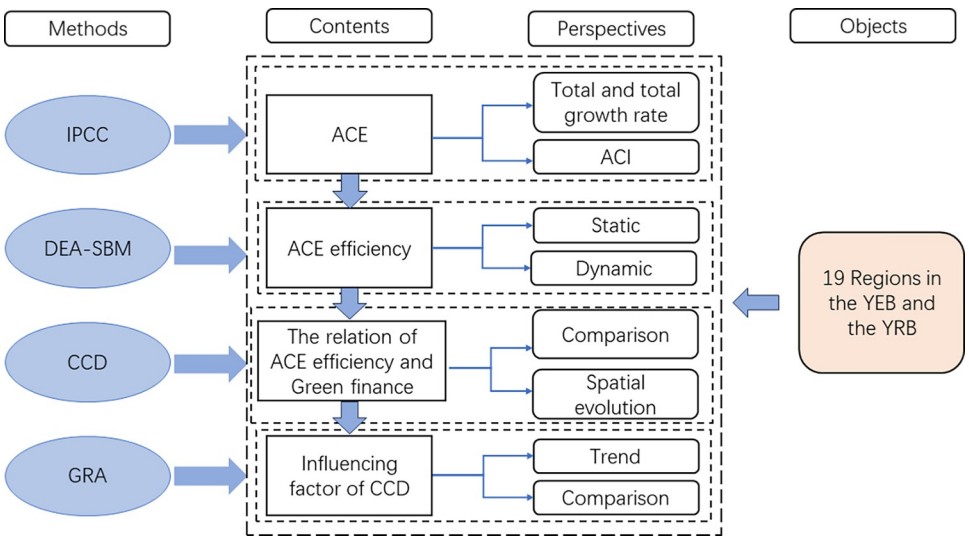

**Fig 1. Theoretical framework.**

## 2.1 Measurement of ACE

When it comes to measuring ACE, the choice of measurement method is the most critical. Referring to the existing research achievements, the methods of carbon emission measurement mainly include the life cycle assessment method [5–8], input-output method [9–13] and IPCC method [14–19]. Among them, the IPCC carbon emission coefficient method has a wide application scale and can be used to calculate carbon emissions at all levels. Its formula is simple, and the principle is easy to understand, so it is the most widely used. For example, [20] calculated China's ACE from 2005 to 2019 using the IPCC carbon emission coefficient method. They found that China's total agricultural carbon emissions showed a downward trend but inter-year fluctuations. The IPCC carbon emission coefficient method was used to preliminarily estimate greenhouse gases from agricultural production in China during 1991–2008, and it was found that methane emissions from planting industry decreased and nitrous oxide emissions increased, the content of methane and nitrous oxide in animal husbandry increased first and then decreased [21]. [22] used the IPCC carbon emission coefficient method to measure the ACE of 31 provinces in China from 1997 to 2017. The research results found that the ACE

**Table 1. Abbreviations of nouns in papers.**

| Noun | Abbreviations |
|---|---|
| Agricultural carbon emissions | ACE |
| Agricultural carbon intensity | ACI |
| Yangtze River Economic Belt | YEB |
| Yellow River Basin | YRB |
| Coupling Coordination Degree | CCD |
| Grey Relation Analysis | GRA |
| Technological progress | TP |
| Technical efficiency | TE |
| Total factor productivity | TFP |
| Data enveloping analysis | DEA |
| Decision-making units | DMUs |

peaked in 2015 and 2016 across the country and in the central and western regions, and the ACE of the eastern region also peaked in 2015. [23] used the IPCC carbon emission coefficient method to measure ACE in Shandong Province from 2000 to 2020, and the research results found regional differences in ACE in 16 cities in Shandong Province, with Heze City ranking first.

## 2.2 ACE efficiency

There are two main approaches to measuring the efficiency of carbon emissions. One is a parametric method, the most commonly used stochastic frontier method [24, 25]. The stochastic frontier method constructs the production frontier through the production function, which possesses statistical characteristics and high stability of output results. However, this method usually only deals with the case of single output, and it is not very easy to deal with the multi-input and multi-output economic system. When there are fewer input indicators, the reliability of the results will be affected due to the correlation between the indicators. The other is non-parametric, mainly represented by data enveloping analysis (DEA) [26–30]. The DEA method constructs the production frontier through a linear programming method. Its advantage is that it can handle the efficiency evaluation of multi-input and multi-output without unifying dimensions, and determining the weights is more objective and more accessible to operate. Therefore, scholars tend to study the ACE efficiency by the DEA method. For example, [31] calculated the change trend of ACE efficiency of China's provinces from 2000 to 2011 based on the DEA method. They found that there were regional differences in ACE efficiency. However, the traditional DEA method can not solve the problem of existing undesirable output and can not compare the decision-making units (DMUs) with the efficiency of 1. In order to make up for the above defects, [32] proposed the SBM model, which was subsequently improved to the SBM model with the undesirable output. Also, it proposed the super-efficient SBM model [33] to compare DMUs with an efficiency of 1. After that, many scholars used the above methods to research. For example, [34] measured the ACE efficiency of 31 provinces in China from 2000 to 2019 based on the undesirable SBM model. [35] measured the ACE efficiency of 30 research objects in China from 2000 to 2019 based on the global SBM model with undesirable output. [36] evaluated the ACE efficiency of 14 cities in Gansu Province using the super-efficiency SBM model with undesirable output. [37] used the super-efficiency SBM method to measure the agricultural ecological efficiency in the middle reaches of the Yangtze River. Recently, based on the super-efficiency SBM models, agricultural ecological efficiency and its influencing factors in China's 31 provinces from 2000 to 2021 are measured by [38]. [39] employed the super-efficiency SBM model to measure the efficiency of new urbanization and eco-efficiency in Fujian Province.

## 2.3 CCD analysis

In exploring the relationship between the two systems involving carbon emission efficiency, some scholars adopted the regression method [40–42], which focused on the unilateral influence of other systems on carbon emission efficiency. Some scholars adopted the decoupling method [43, 44], which mainly studied the interaction between economic development and carbon emissions. Some scholars adopted CCD analysis, which focused on the degree of interdependence and mutual restriction between carbon emission efficiency and other systems and focused on the CCD analysis between carbon emission efficiency and economic development, industrial structure optimization, and urbanization. For instance, [20] explored the CCD between ACE efficiency and agricultural economic growth from the provincial perspective in China. [45] evaluated and analyzed the CCD from the perspective of time series and space for

the binary and ternary systems composed of carbon emission, economic development, and environmental protection in various provinces and cities in China. Using the CCD model, [46] investigated the relationship between carbon emission efficiency and industrial structure optimization in six central provinces. [47] used the CCD model to couple the industrial carbon emission efficiency with the rationalization and upgrading of industrial structure of provinces along the YEB, respectively. [48] explored the CCD between new urbanization and ACE efficiency in 30 provinces. [49] explored the CCD between the carbon emission efficiency of the construction industry and urbanization in Hebei Province. In analyzing the two systems, there needs to be more literature to explore the dependent and constrained relationship between ACE efficiency and green finance. [50] employed CCD model to evaluate the degree of the coupling and coordinated development of the digital economy and carbon emission efficiency in provincial regions of China. This paper adopts the CCD method.

## 2.4 Influencing factors analysis

At present, when exploring the influencing factors of CCD, some scholars used Tobit regression, such as [51] analyzed the factors affecting the CCD of environment and economy by Tobit regression model, and [52] constructed a Tobit regression model to verify the influence degree of various factors on the CCD of digitization and higher education in China. Some scholars used spatial econometric models to explore the influencing factors of the CCD. For example, [53] used the spatial Durbin model to explore the influencing factors of the CCD between rural population structure and financial resource allocation, and [54] used the geographically weighted model to explore the influencing factors of the CCD between production, life, and ecological space. Some scholars used GRA to explore the influencing factors of CCD, such as [55] used GRA to explore the influencing factors of the CCD of green urbanization in the Yangtze River Delta. [56] used GRA to explore the influencing factors of the CCD of new urbanization and rural revitalization. [57] used GRA to explore the influencing factors of the CCD of technological innovation and standardization. The GRA method is a method to measure the degree of correlation between factors according to the similarity or dissimilarity of the development trend between factors. Compared with Tobit regression and spatial econometric analysis methods, GRA has the advantage that there is no excessive requirement on the sample size, it does not require typical distribution laws, and its computation is relatively tiny. It is a relatively simple and reliable method of system analysis. Because the sample size in this paper is small, and there is no typical distribution law. The results of the analysis using the GRA method are more reliable.

## 2.5 Summary and contribution

The above literatures have deeply discussed the agricultural carbon emission from different perspectives, covering the sources and measurement of agricultural carbon emission, the measurement of agricultural carbon emission efficiency, the relationship between the two systems involved in carbon emission efficiency, and the influencing factors of coupling coordination degree. However, there are some limitations in the existing researches: (1) There are few comparative analyses of agricultural carbon emission efficiency in the two basins in the existing literatures.(2) Although there are studies on the relationship between agricultural carbon emission and green finance in existing literatures, few literatures have studied the coupling coordination degree between agricultural carbon emission efficiency and green finance and the influencing factors of the coupling coordination degree. Based on the above literature studies, this paper firstly uses the IPCC carbon emission coefficient method to measure ACE and ACI in various regions of the YEB and YRB, then applies the super-efficiency SBM model to

obtain the ACE efficiency of each study region of the YEB and YRB, and then analyzes the CCD of ACE efficiency and green finance. Finally, the GRA method is used to study further the influencing factors of the CCD of the two systems.

Different from the existing research literature, the contributions of this paper are as follows: (1) Since the Yangtze River and Yellow River basins are extensive in scope and have certain geographical environmental characteristics, it is better to take the basin as the research object than the regional study of administrative division, so as to better study the carbon emission under the difference of physical geographical environment, analyze the difference between the north and the south from the perspective of the basin, and provide targeted suggestions for the development of different regions.(2) Existing literature studies focus on the coupling and coordination analysis of carbon emission efficiency, economic development and industrial structure optimization. This paper innovatively analyzes the coupling and coordination analysis of carbon emission efficiency and green finance.

## 3. Methods

Firstly, the measurement method of ACE and ACI is briefly introduced, and then the Super-efficiency SBM model and Malmquist index are derived. Next, entropy method and coupling coordination degree method are presented. Finally, we give rise to gray relation analysis method.

### 3.1 Measurement of ACE and ACI

In this paper, ACE is regarded as an undesirable output. Since the existing data cannot be directly obtained from the data of ACE, it is necessary to obtain it through calculation. The IPCC carbon emission coefficient method calculates ACE in different regions. The calculation formula is as follows:

$$C = \sum C_i = \sum T_i \times \delta_i \tag{1}$$

Where $C$ represents the total ACE, $C_i$ represents the carbon emissions of various carbon sources, $T_i$ represents the quantity of various carbon sources, and $\delta_i$ represents the carbon emission coefficient corresponding to various carbon sources. Concerning the research results of [34] and [21], combined with the characteristics of ACE in different regions and the availability of data, the ACE is calculated from three aspects: agricultural materials input, rice planting, and animal husbandry.

ACE in agricultural materials input includes chemical fertilizers, pesticides, agricultural plastic films, diesel oil, farmland plowing (expressed by the land sown area), and farmland irrigation. The carbon emission coefficients of the above carbon sources are shown in Table 2.

Rice planting mainly produces methane. Due to the differences in the carbon emission coefficient of rice in different regions, the carbon emission coefficient of rice planting in

**Table 2. Carbon emission resources and coefficient of agricultural inputs.**

| Carbon Emission Resources | Carbon Emission Coefficients |
|---|---|
| Fertilizer | 0.8956kg(c)/kg |
| Pesticide | 4.9341kg(c)/kg |
| Agricultural plastic film | 5.18kg(c)/kg |
| Diesel oil | 0.5927kg(c)/kg |
| Farmland irrigation | 266.48kg(c)/hm2 |
| Farmland ploughing | 312.6kg(c)/hm2 |

**Table 3. Carbon emission resources and coefficient of rice cultivation.**

| Carbon Emission Resources | Carbon Emission Coefficients |
|---|---|
| Qinghai (QH) | 0 kg(CH4)/hm$^2$ |
| Gansu (GS) | 668.3 kg(CH4)/hm$^2$ |
| Ningxia (NX) | 73.5 kg (CH4)/hm$^2$ |
| Inner Mongolia (IM) | 89.3 kg(CH4)/hm$^2$ |
| Henan (HA) | 178.5 kg(CH4)/hm$^2$ |
| Shaanxi (SN) | 125.1 kg(CH4)/hm$^2$ |
| Shanxi (SX) | 66.2 kg(CH4)/hm$^2$ |
| Shandong (SD) | 210 kg(CH4)/hm$^2$ |
| Sichuan (SC) | 169.3 kg(CH4)/hm$^2$ |
| Chongqing (CQ) | 169.3 kg(CH4)/hm$^2$ |
| Guizhou (GZ) | 160.5 kg(CH4)/hm$^2$ |
| Yunnan (YN) | 57.4 kg(CH4)/hm$^2$ |
| Hubei (HB) | 382.3 kg(CH4)/hm$^2$ |
| Hunan (HN) | 350.3 kg(CH4)/hm$^2$ |
| Anhui (AH) | 318.6 kg(CH4)/hm$^2$ |
| Jiangxi (JX) | 422.3 kg(CH4)/hm$^2$ |
| Jiangsu (SX) | 324.1 kg(CH4)/hm$^2$ |
| Zhejiang (ZJ) | 356.1 kg(CH4)/hm$^2$ |
| Shanghai (SH) | 312.6 kg(CH4)/hm$^2$ |

different regions was determined concerning the research of [21], as shown in Table 3. Since the carbon produced by rice planting involves methane emission, to facilitate subsequent analysis, 1 ton of methane is equivalent to 6.82 tons of carbon by referring to the results of the fourth assessment of IPCC.

Animal husbandry includes the carbon emissions from intestinal fermentation and cattle, sheep, pigs, and other livestock manure. The carbon emission coefficients of the above carbon sources are shown in Table 4.

ACI refers to the carbon emissions generated per unit of gross agricultural product, reflecting the relationship between regional ACE and regional economic development. The formula is:

$$C_{1t} = C_t / GDP_t \tag{2}$$

where $C_{1t}$ represents the ACI in a certain period, $C_t$ represents the ACE in a certain period, $GDP_t$ represents the gross agricultural product in a certain period.

## 3.2 Super-efficiency SBM model

In 1978, [58] proposed a non-parametric method of DEA to calculate the efficiency of multiple DMUs, which was later used in performance evaluation. DEA method uses the linear programming method to determine the production frontier from the perspective of input and

**Table 4. Carbon emission resources and coefficient of animal husbandry.**

| Carbon Emission Resources | Carbon Emission Coefficients |
|---|---|
| Pigs | 34.091kg(c)/(head·year) |
| cattle | 415.91 kg(c)/ (head·year) |
| sheep | 35.182 kg(c)/ (head·year) |

output. It measures the efficiency according to the degree to which the DMUs deviate from the production frontier. The traditional DEA methods represented by CCR and BCC measure the efficiency of DMUs based on radial and angular aspects. The undesirable output SBM model can solve the problem of slack variables, and the super-efficient SBM model can sort effective DMUs. Therefore, this paper's final efficiency evaluation model is the super-efficient SBM model containing the undesirable output.

Suppose there are $n$ decision making units that need to evaluate ACE efficiency, denoted as $DMU_j(j = 1,2,\ldots,n)$, each decision unit has $m$ inputs, denoted as $X_i(i = 1,2,\ldots,m)$, each decision unit has $q_1$ desirable outputs, denoted as $y_r(r = 1,2,\ldots,q_1)$, each decision unit has $q_2$ undesirable outputs, denoted as $b_t(t = 1,2,\ldots,q_2)$, and the final super-efficiency SBM model containing undesirable outputs is as follows:

$$\min\rho = \frac{1 + \frac{1}{m}\sum_{i=1}^{m} s_i^- / x_{ik}}{1 - \frac{1}{q_1+q_2}\left(\sum_{r=1}^{q_1} s_r^+ / y_{ik} + \sum_{t=\cdot}^{q_2} s_t^{b-} / b_{rk}\right)} \tag{3}$$

$$s.t. \begin{cases} \sum_{j=1,j\neq k}^{n} x_{ij}\lambda_j - s_i^- \leq x_{ik} \\[2mm] \sum_{j=1,j\neq k}^{n} y_{rj}\lambda_j + s_r^- \geq y_{rk} \\[2mm] \sum_{j=1,j\neq k}^{n} b_{tj}\lambda_j - s_t^{b-} \leq b_{tk} \\[2mm] 1 - \frac{1}{q_1 + q_2}\left(\sum_{r=1}^{q_1} s_r^+ / y_{ik} + \sum_{t=1}^{q_2} s_t^{b-} / b_{rk}\right) > 0 \end{cases} \tag{4}$$

Where $\lambda, s^-, s^+ \geq 0; i = 1, 2, \ldots, m; r = 1, 2, \ldots, q; j = 1, 2, \ldots, n(j \neq k)$.

### 3.3 Malmquist index

In this paper, the Malmquist index was used for dynamic analysis of panel data to observe the dynamic change trend of ACE efficiency in different regions during the research period. The Malmquist index from period t to period t+1 is as follows:

$$M_{TFP}(x_{t+1}, y_{t+1}, x_t, y_t) = \sqrt{\frac{D^t(x_{t+1}, y_{t+1})}{D^t(x_t, y_t)} \times \frac{D^{t+1}(x_{t+1}, y_{t+1})}{D^{t+1}(x_t, y_t)}} \tag{5}$$

The Malmquist index can be broken down into technical efficiency (TE) and technological progress (TP):

$$M_{TFP}(x_{t+1}, y_{t+1}, x_t, y_t) = \frac{D^{t+1}(x_{t+1}, y_{t+1})}{D^t(x_t, y_t)} \times \sqrt{\frac{D^t(x_{t+1}, y_{t+1})}{D^{t+1}(x_{t+1}, y_{t+1})} \times \frac{D^t(x_t, y_t)}{D^{t+1}(x_t, y_t)}} = TE \times TP \tag{6}$$

When TFP>1, it proves that the total factor productivity has improved. When TFP = 1, the total factor productivity does not change. When TFP<1, it indicates that total factor productivity has decreased. TE>1 indicates that the technical efficiency is improved, and the gap between the DMUs and the efficiency frontier is narrowed. When TE = 1, it indicates that the technical efficiency is unchanged. TE<1 indicates a decrease in technical efficiency and widens

the gap between the DMUs and the efficiency frontier. When TP>1, it indicates the progress of technology and innovation and the expansion of the production boundary. When TP = 1, it indicates no change in technology and innovation. When TP<1, it indicates that technology and innovation have regressed and production boundary has shrunk.

## 3.4 Entropy method

Since the entropy method starts from data, it can objectively reflect the importance of indicators. In this paper, the entropy method is used to calculate the weight of each indicator on the pre-processed data, and then the development level of green finance in various regions can be calculated.

The specific steps and formulas for calculating the weight of each index are as follows:

1. Calculate the proportion of the jth index in the ith evaluated object:

$$R_{ij} = X'_{ij} / \sum_{i=1}^{m} X'_{ij} \tag{7}$$

2. Calculate the entropy of the index:

$$e_j = \frac{-1}{\ln m} \times \sum_{i=1}^{m} R_{ij} / \ln R_{ij} \tag{8}$$

3. Calculate weights:

$$w_j = \frac{g_j}{\sum_{j=1}^{n} g_j}, g_j = 1 - e_j \tag{9}$$

4. Calculate the development level of green finance for object ith:

$$U_i = \sum_{i=1}^{m} w_j \times X'_{ij} \tag{10}$$

## 3.5 Coupling coordination degree

The coupling degree model describes the correlation degree between two or more systems. The coupling degree model of ACE efficiency and green finance level follows:

$$W = 2\sqrt{X_1 X_2} / (X_1 + X_2) \tag{11}$$

Where $W$ represents the coupling degree of the two systems, $X_1$ represents the level of agricultural carbon emission system, measured by agricultural carbon emission efficiency, and $X_2$ represents the level of green finance system, measured by the development level of green finance. The value of coupling degree $W$ ranges from 0 to 1. The closer the coupling coordination degree is to 0, the lower the degree of correlation between the two systems is proved, and the closer it is to 1, the higher the degree of correlation is proved. That is, the higher the correlation between agricultural carbon emission efficiency and green finance development level.

However, some things could be improved in the coupling degree model, which cannot reflect the coordinated development of the two systems. For example, when ACE efficiency and green finance development levels are the same and relatively small, the coupling degree will be close to 1, ignoring the coordinated development between the two systems. Therefore, in order to better analyze the coupling coordination relationship between the two systems, the CCD model is introduced:

$$S = \sqrt{W \times T}, T = \alpha X_1 + \beta X_2 \qquad (12)$$

Where S represents the CCD of agricultural carbon emission efficiency and green finance development level, $T$ represents the comprehensive evaluation index of the two system levels, $\alpha$、 $\beta$ is the undetermined coefficient and $\alpha + \beta = 1$ reflects the contribution degree of the two parts to the overall development respectively. In this paper, the importance of the two systems is the same, and $\alpha$、 $\beta$ are 0.5. The value range of CCD is also 0–1, and the closer it is to 1, the better the coupling coordination development of the two systems. In order to facilitate classification and analysis, given the existing literature (Qian et al. 2016), the CCD is classified into grades. Divided into ten stages: extreme disorder ($0 \leq CCD \leq 0.1$), serious disorder ($0.1 < CCD \leq 0.2$), moderate disorder ($0.2 < CCD \leq 0.3$), mild disorder ($0.3 < CCD \leq 0.4$), near disorder ($0.4 < CCD \leq 0.5$), barely coordinated ($0.5 < CCD \leq 0.6$), primary coordination ($0.6 < CCD \leq 0.7$), intermediate coordination ($0.7 < CCD \leq 0.8$), good coordination ($0.8 < CCD \leq 0.9$), high-quality coordination ($0.9 < CCD \leq 1.0$).

### 3.6 Gray relation analysis

This paper uses GRA to analyze the influencing factors of the CCD of the two systems. GRA refers to the relative strength of dependent variables affected by other factors in a grey system.

The steps of GRA are as follows:

1. Non-dimensional processing. The sequence of CCD and each influencing factor after dimensionless processing by initializing method are denoted as $X_0'(n, t)$ and $X_i'(n, t)$, where $i = 1,2,3,4$ represents the four influencing factors studied respectively, n represents the nth object, and t represents the t year.

2. Calculate the relation coefficient:

$$r_i(n, t) = \frac{\min|X_0'(n, t) - X_i'(n, t)| + \rho \max|X_0'(n, t) - X_i'(n, t)|}{|X_0'(n, t) - X_i'(n, t)| + \rho \max|X_0'(n, t) - X_i'(n, t)|} \qquad (13)$$

where $\rho = 0.5$.

3. Calculate the GRA of each region:

$$r_{i,n} = \frac{1}{T} \sum_{t=1}^{T} r_i(n, t) \qquad (14)$$

## 4. Data sources and indicator selections

### 4.1 ACE efficiency

This paper uses the data of 19 regions in China from 2001 to 2020 (including 8 regions in the YRB and 11 regions in the YEB); according to the Outline of the YEB Development Plan, the YEB includes 11 regions: Sichuan, Chongqing, Guizhou, Yunnan, Hubei, Hunan, Anhui,

**Table 5. Input-output index for measuring agricultural carbon emission efficiency.**

| Index | Indicator Type | Specific Index |
|---|---|---|
| Input index | Labor input | Number of employees in the primary industry |
| | Land input | sown area of crops |
| | Agricultural materials input | chemical fertilizer application |
| | | pesticide application amount |
| | | Agricultural film usage |
| | | Agricultural diesel usage |
| Output index | Desirable output | the total output value of agriculture and animal husbandry |
| | Undesirable output | ACE |

Jiangxi, Jiangsu, Zhejiang, and Shanghai. Since the YRB only flows through Aba Prefecture and Ganzi Prefecture of Sichuan Province, it has little impact on the overall ACE of the YRB, and the Outline of the YEB Development Plan approved by The State Council includes Sichuan Province in the YEB. Given the above considerations, in this paper, Qinghai, Gansu, Ningxia, Inner Mongolia, Henan, Shaanxi, Shanxi and Shandong are divided into the YRB.

Based on previous research results [34, 35, 59], this paper considers input indicators from the three levels of labor, land, and agricultural materials, takes the total output value of agriculture and animal husbandry as the desirable output, and ACE as the undesirable output, and constructs an input-output index system, as shown in Table 5.

The specific analysis is as follows:

Input index: Labor input is measured by the number of employees in the primary industry, the sown area of crops measures land input, and agricultural materials input includes chemical fertilizer, pesticide, agricultural film, and agricultural diesel, respectively, by the conversion amount of chemical fertilizer application, pesticide application amount, agricultural film usage and agricultural diesel usage.

Output index: To eliminate the interference of price factors, the total output value of agriculture and animal husbandry is treated at a constant price, with 2001 as the base year. Since the existing data cannot directly obtain the data on agricultural carbon emissions, it is necessary to obtain the data through measuring, see section 2.1 of the article.

The data are derived from the China Statistical Yearbook, China Rural Statistical Yearbook, and the statistical yearbook of each province and city.

## 4.2 Development level of green finance

In order to measure the development level of green finance in the YEB and the YRB, referring to the existing research results of green finance [60, 61], the following index system is constructed based on the scientific nature of the index system construction and the availability of data and the development level of green finance is measured from five dimensions, as shown in Table 6. Since the "Opinions on Implementing Environmental Protection Policies and Regulations to Prevent Credit Risks" promulgated in 2007 is generally regarded as the beginning of China's green finance practice, the research period is from 2007 to 2020, and the data are from China Statistical Yearbook and choice database.

In order to ensure the research results are accurate due to the significant difference in the index values, this paper adopts the method of range standardization to process the data. For the forward index, the calculation formula is as follows:

$$X'_{ij} = (X_{ij} - X_{jmin})/(X_{jmax} - X_{jmin}) \tag{15}$$

**Table 6. Index system of green finance development level.**

| Target layer | System layer | Index layer and unit | Weight | Index properties |
|---|---|---|---|---|
| Green finance | Green credit | Total interest expenses of six energy-intensive Industries /Total financial expenses of industrial industries (%) | 0.0452 | Inverse index |
| | Green investment | Industrial pollution control investment /GDP(%) | 0.1842 | Forward index |
| | Green insurance | Agricultural insurance income/Total agricultural output value (%) | 0.2462 | Forward index |
| | Green securities | Total turnover of environmental protection enterprises/Total turnover of A-shares (%) | 0.4735 | Forward index |
| | Government environmental protection support | Environmental protection expenditure/General Budget Expenditure (%) | 0.0509 | Forward index |

For inverse index, the calculation formula is as follows:

$$X'_{ij} = (X_{ij} - X_{j\min})/(X_{j\max} - X_{j\min}) \tag{16}$$

Where $X_{ij}$ is the value of index j in the ith evaluated object, $X_{j\min}$ and $X_{j\max}$ are the minimum and maximum values of index j in each sample respectively, and $X'_{ij}$ is the standardized value of index j.

## 4.3 Influencing factors of CCD

Concerning the existing research results [55, 62], this paper finally determined the following influencing factors: (1) economic development level: The level of economic development is the material basis for promoting the development of green finance and improving the efficiency of ACE. Economic development can provide sufficient funds for scientific research, education, and infrastructure construction for various regions' green finance industries and the agricultural sector. In this paper, the per capita GDP of each region is used as an index to measure the level of economic development. (2) Government regulation and control: The government's macro-regulation and control can incline resources to industries that promote the development of the whole society. This paper uses per capita regional financial expenditure as a measure index. (3) Quality of human capital: High-quality human resources are more likely to adopt advanced agricultural technologies and management methods to reduce agricultural carbon emissions and improve agricultural carbon emission efficiency. At the same time, the promotion and application of green finance requires certain professional knowledge and skills. By improving the quality and ability of agricultural producers, it can promote the transformation and upgrading of agricultural production mode, realize the green and low-carbon agricultural production, and promote the promotion and application of green finance in the agricultural field, and promote the coupling and coordinated development of agricultural carbon emission efficiency and green finance. This paper measures the proportion of the population aged 6 and above who have received a high school education. (4) Scientific and technological innovation ability: Scientific and technological innovation is the core of improving the competitiveness of green finance, and the use of innovative technologies in pollution control, energy conservation, and emission reduction is conducive to the growth of green performance. Scientific and technological innovation can promote the transformation of agricultural production mode, promote the development of green industry, and then realize the coordinated development of agricultural carbon emission efficiency and green finance. At the same time, scientific and technological innovation capacity can also improve the efficiency and quality of green financial products and services, and enhance their supporting role in

agricultural carbon emission efficiency. In this paper, R&D expenditure is taken as a measure of scientific and technological innovation ability.

# 5. Results and analysis

## 5.1 Statistical analysis of ACE and ACI

**5.1.1 Total agricultural carbon emissions.** The IPCC carbon emission coefficient method measures ACE in various regions. Based on the index system established above, the collected data is brought into Formula (1) to calculate the ACE of the YEB and the YRB from 2001 to 2020. The origin software is applied to visualize the results and draw Figs 2 and 3.

The comparison between Figs 2 and 3 shows that the total ACE in the YEB is far higher than that in the YRB. From 2001 to 2020, the total ACE in the YEB remained between 100–125 million tons, with an average of 11.41 million tons, while the total ACE in the YRB remained between 50–70 million tons. The average is 59.72 million tons. In terms of total and average ACE, the YEB is almost twice that of the YRB. From the perspective of change trend, the total

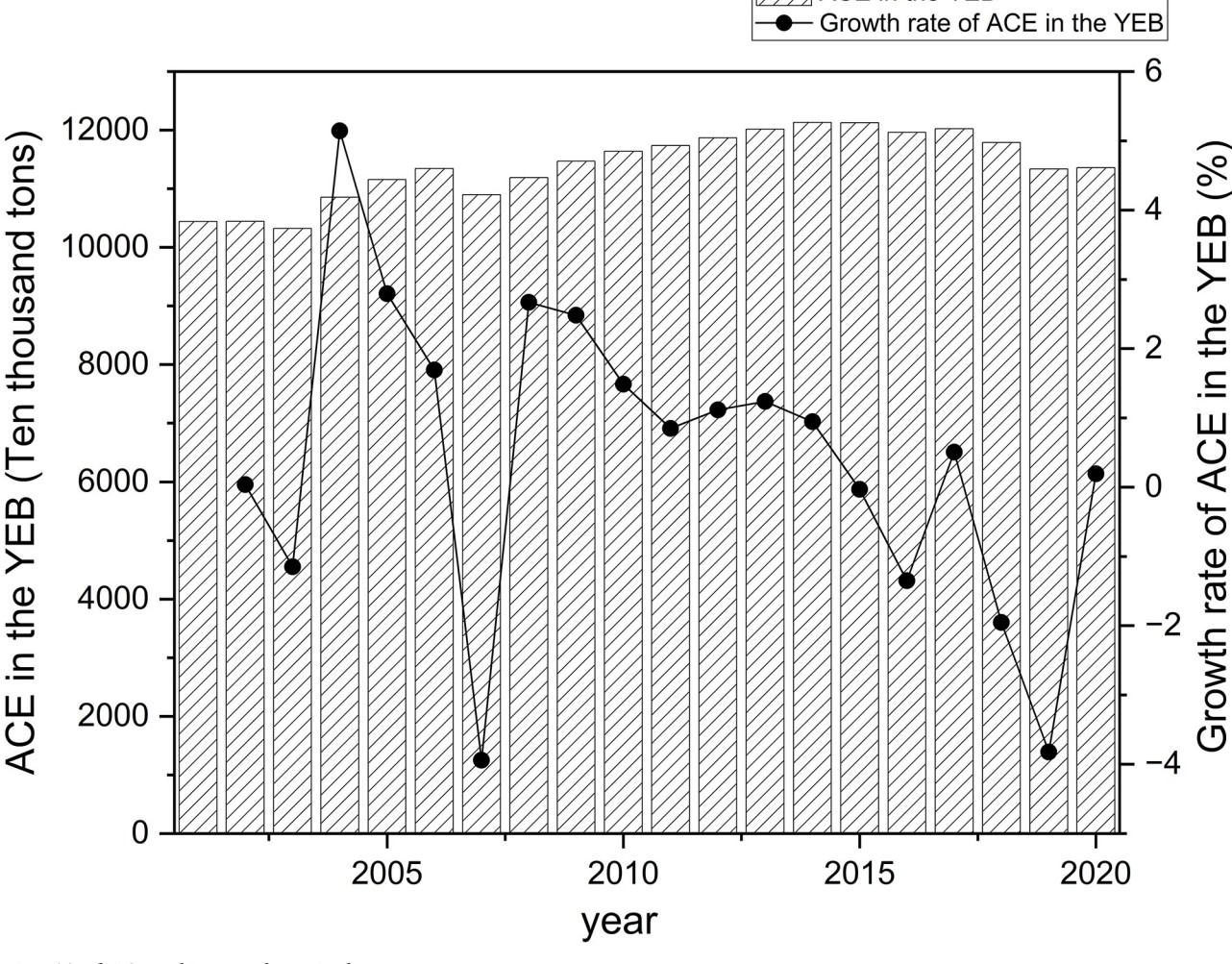

**Fig 2. Total ACE and its growth rate in the YEB.**

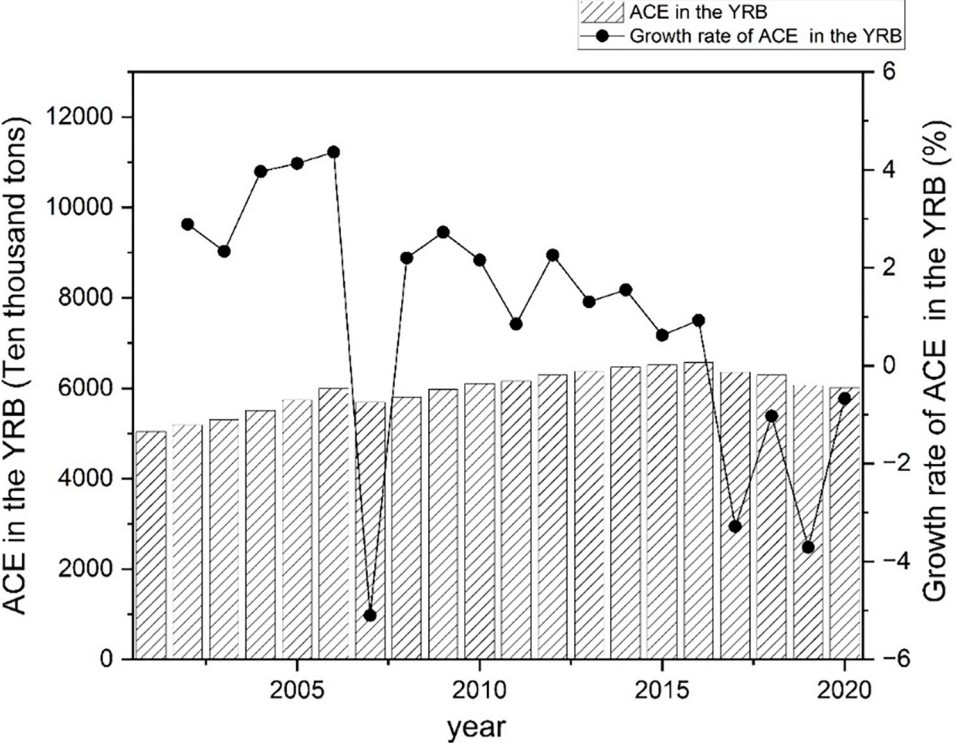

**Fig 3. Total ACE and its growth rate in the YRB.**

ACE in the YEB and the YRB show roughly "inverted U-shaped," and the YEB and the YRB reached their peaks in 2014 and 2016.

Regarding the growth rate of total ACE, the YEB and the YRB mainly have favorable growth rates before the peaks. However, the growth rates gradually approach 0, reflecting that the growth rates of ACE are getting smaller and smaller, and the growth rates show fluctuating trends after the peaks. From the general trend, although the growth rate of ACE in both the YEB and the YRB show a trend of fluctuation and decline, the growth rate of total ACE in the YEB decreases faster than that in the YRB. From 2002 to 2006, the total ACE in the YRB over-grew, while the YEB increased in fluctuation, reaching its lowest point in 2007. After 2007, the growth rate fluctuated, and the decline rate of the YEB was faster. There is the nodal time of growth rate change in 2007, related to policies such as the "Comprehensive Work Plan for Energy Conservation and Emission Reduction" promulgated by the National Development and Reform Commission. After that, China began to pay close attention to the responsible implementation of energy conservation and emission reduction. Law enforcement supervision established a robust leadership coordination mechanism for energy conservation and emission reduction, achieving apparent results in energy conservation and emission reduction.

**5.1.2 Agricultural carbon emission intensity.** ACI in the YEB and the YRB are calculated according to the ACE calculated in Section 2.1 and the gross agricultural and animal husbandry production in the China Rural Statistical Yearbook. Origin software is applied to visualize the calculation results, and the results are shown in Fig 4.

From 2001 to 2020, the ACI of the YEB is higher than that of the YRB, from 12.33 tons/ ten thousand yuan in 2001 to 2.90 tons /ten thousand yuan in 2020 in the YEB, and from 8.31 tons / ten thousand yuan in 2001 to 1.90 tons / ten thousand yuan in 2020 in the YRB. The ACI in

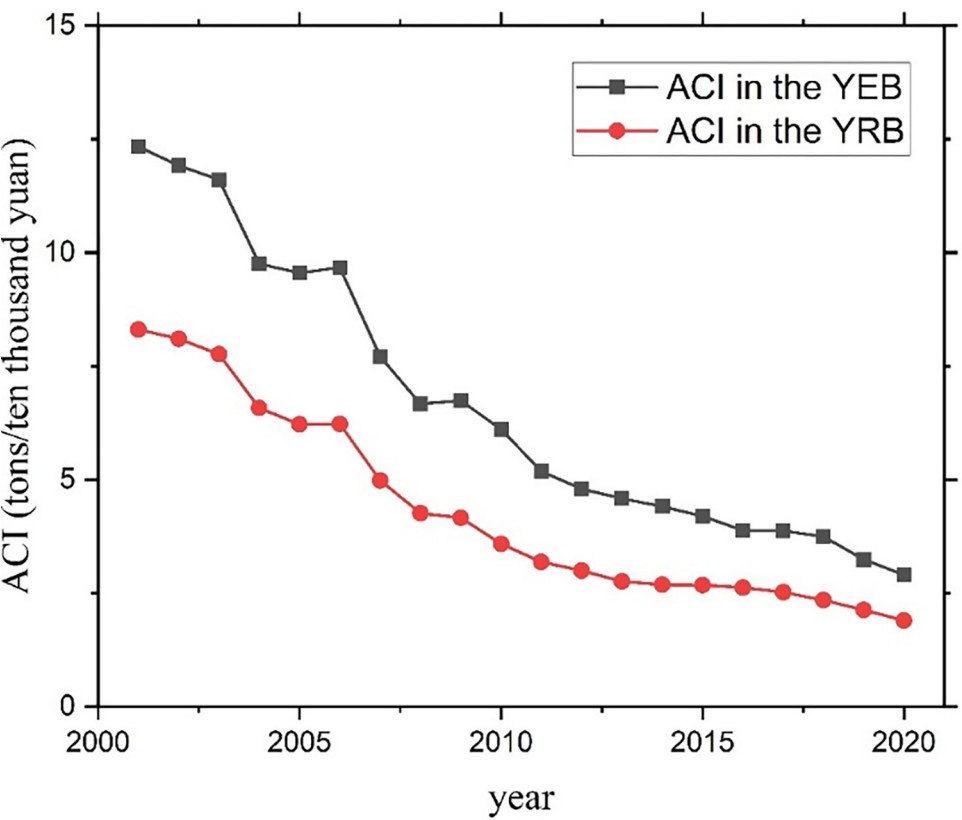

**Fig 4. Comparison chart of ACI in YEB and YRB.**

the YEB and the YRB both show downward trends. The ACI in the YEB in 2020 is 23.53% of that in 2001, and that in the YRB in 2020 is 22.84% of that in 2001, indicating that the decline rate of the ACI in the YRB is faster. However, the difference in ACI between the YEB and the YRB is getting smaller and smaller. In terms of total ACE, the total ACE in the YEB and the YRB both rise first and then decline. However, the ACI both show downward trends, indicating that gross agricultural and animal husbandry production growth rates are more significant than ACE's. It can be seen that the agricultural production and lifestyles in the YEB and the YRB are constantly developing in a good trend, the agricultural and industrial structure is constantly optimized, the scientific and technological strength and comprehensive production capacity are constantly strengthened, and the standardization and industrialization of agriculture are constantly improved.

## 5.2 Agricultural carbon emission efficiency

**5.2.1 Static analysis of the super-efficient SBM model.** Based on the input-output index system established in Section 4.1, this part uses the super-efficiency SBM model to calculate the ACE efficiency in the YEB and the YRB from 2001 to 2020. In order to facilitate the observation of the evolution process of ACE efficiency in the YEB and the YRB, the annual average values of the YEB and the YRB included areas are taken as the efficiency values of the YEB and the YRB, and the obtained results are visualized with origin software, as shown in Fig 5.

The ACE efficiencies of both the YEB and the YRB show overall increasing trends from 2001 to 2020. The ACE efficiency of the YEB reached 0.83 in 2020, a significant improvement

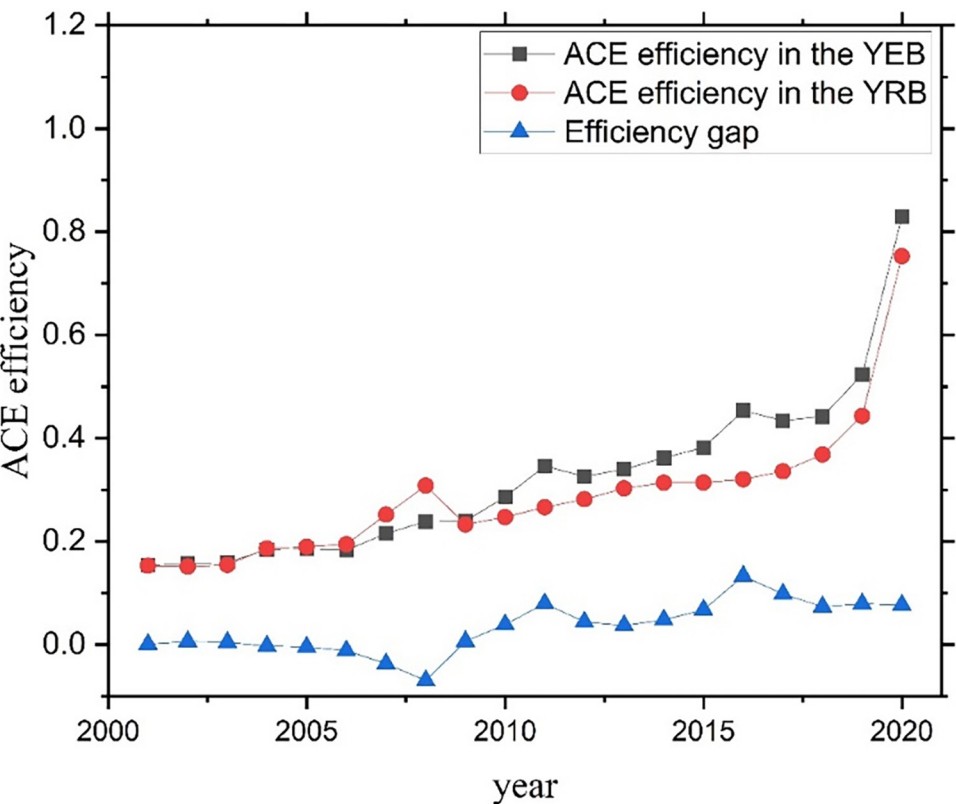

**Fig 5. ACE efficiencies in YEB and YRB.**

from 0.15 in 2001. The ACE efficiency of the YRB increased from 0.15 in 2001 to 0.75 in 2020. From 2001 to 2005, the ACE efficiency of the YEB was similar to that of the YRB, and from 2006 to 2008, the ACE efficiency of the YRB was higher than that of the YRB. In 2009, the ACE efficiency of the YEB was higher than that of the YRB, and the efficiency gap reached the largest in 2016. There is little difference in ACE efficiency between the YEB and the YRB, and the YRB is slightly higher. This shows that after the state implemented a series of policies to protect the ecological environment, the agricultural sector in the YEB and the YRB has achieved excellent results in terms of carbon emissions, creating conditions for promoting the high-quality development of rural agriculture under low-carbon agriculture and helping to realize better the goals of "carbon neutrality" and "carbon peak" Based on the input-output index system established in Section 4.1, this part uses the super-efficiency SBM model to calculate the ACE efficiency in the YEB and the YRB from 2001 to 2020. In order to facilitate the observation of the evolution process of ACE efficiency in the YEB and the YRB, the annual average values of the YEB and the YRB included areas are taken as the efficiency values of the YEB and the YRB, and the obtained results are visualized with origin software, as shown in Fig 5.

In order to more clearly observe the change process of ACE efficiency in various regions of the YEB and YRB, the ACE efficiency values in various regions of the YEB and YRB are divided into four periods from 2001 to 2005, from 2006 to 2010, from 2011 to 2015 and from 2016 to 2020. The ACE efficiencies are divided into five levels: 0–0.2, 0.2–0.4, 0.4–0.6, 0.6–0.8, greater than 0.8, respectively, representing ACE efficiencies that are very low, low, medium, high, and very high. The carbon emission efficiencies of each region of the YEB and YRB in four periods are further calculated, and the calculation results are shown in Tables 7 and 8.

**Table 7. ACE efficiencies in different regions of the YEB.**

| Regions | Average Efficiency 2001–2005 | Grade | Average Efficiency 2006–2010 | Grade | Average Efficiency 2011–2015 | Grade | Average Efficiency 2016–2020 | Grade |
|---|---|---|---|---|---|---|---|---|
| SC | 0.1916 | very low | 0.3266 | low | 0.4585 | medium | 1.0279 | very high |
| CQ | 0.1553 | very low | 0.2214 | low | 0.3150 | low | 1.0073 | very high |
| GZ | 0.1670 | very low | 0.2203 | low | 0.4832 | medium | 1.1025 | very high |
| YN | 0.1129 | very low | 0.1592 | very low | 0.2444 | low | 0.6133 | high |
| HB | 0.1463 | very low | 0.2566 | low | 0.3354 | low | 1.0170 | very high |
| HN | 0.1799 | very low | 0.2739 | low | 0.2879 | low | 1.0298 | very high |
| AH | 0.1221 | very low | 0.1888 | very low | 0.2316 | low | 0.3513 | low |
| JX | 0.1431 | very low | 0.2021 | low | 0.2616 | low | 0.4432 | medium |
| JS | 0.1776 | very low | 0.2957 | low | 0.5206 | medium | 1.0162 | very high |
| ZJ | 0.1838 | very low | 0.3290 | low | 0.4610 | medium | 1.0571 | very high |
| SH | 0.2693 | low | 0.6747 | high | 0.6035 | high | 0.4587 | medium |

The ACE efficiencies of the YEB and the YRB have been increasing with time, and the growth degree of different regions is quite different. In the YEB, Anhui has the smallest increase in ACE efficiency, and all four periods are at the lowest levels. Anhui is a central agricultural province in the YEB. As high output is accompanied by high carbon emission, and there is much room for improvement in the degree of agricultural intensification and technical level, the efficiency of ACE is relatively low. The growth rate of Zhejiang is the largest, from the lowest level in 2001–2005 to the highest level in 2015–2020, while the growth rate of Sichuan, Guizhou, and Jiangsu are also relatively large. Among them, Zhejiang and Jiangsu, located in the lower reaches of the YEB, have unique geographical locations, a high degree of agricultural intensification, and a technical level, so ACE efficiency is high. As the upper reaches of the YEB, Sichuan, and Guizhou have higher agricultural ACE efficiency because they have adjusted their industrial structure according to local conditions, vigorously planted cash crops, developed specialized agriculture, and promoted the cultivation of cash crops such as fruits,

**Table 8. ACE efficiencies in different regions of the YRB.**

| Regions | Average Efficiency 2001–2005 | Grade | Average Efficiency 2006–2010 | Grade | Average Efficiency 2011–2015 | Grade | Average Efficiency 2016–2020 | Grade |
|---|---|---|---|---|---|---|---|---|
| QH | 0.4035 | medium | 0.6111 | very high | 0.3896 | low | 0.6468 | high |
| NX | 0.0953 | very low | 0.1138 | very low | 0.1371 | very low | 0.2178 | low |
| GS | 0.1189 | very low | 0.1671 | very low | 0.2481 | low | 0.4848 | medium |
| IM | 0.1676 | very low | 0.2258 | low | 0.2915 | low | 0.3910 | low |
| HA | 0.1625 | very low | 0.2303 | low | 0.3134 | low | 0.5099 | medium |
| SN | 0.1445 | very low | 0.2450 | low | 0.4212 | medium | 0.6520 | high |
| SX | 0.0979 | very low | 0.1457 | very low | 0.2200 | low | 0.2596 | low |
| SD | 0.1465 | very low | 0.2374 | low | 0.3456 | low | 0.3921 | low |

tea, cotton, and Chinese medicinal herbs. The situation of agricultural development in the YEB is different, the situation in the coastal areas is better, and the central agricultural provinces and cities have significant room for progress.

In the YRB, the ACE efficiency of Gansu is the smallest, and all four periods are at the lowest level. The reasons are that the natural environment needs to be corrected, the economic strength is backward, the agricultural technology needs to be more robust, and there needs to be a complete and reasonable agricultural industrial structure according to local conditions. Shaanxi has the most significant growth rate, from the lowest level at the beginning to the higher level at the end. Although there is no good natural environment, Shaanxi adjusts its agricultural industrial structure according to local conditions, gives full play to the resource advantages of various regions, and increases the sowing of cash crops while reducing the sowing of food crops. Low-efficiency areas can learn from high-efficiency areas with similar natural geographical conditions, learn from their agricultural policies, and develop agriculture in combination with the characteristics of the region so as to move faster and better to the high-efficiency level.

**5.2.2 Dynamic analysis of the Malmquist index.** Based on the input-output index system established in Section 4.1, this part uses MATLAB software to measure the total factor productivity, technical efficiency index, and technological progress index of 19 regions in the YEB and YRB, and the results are shown in Table 9.

From the perspective of mean value, the TFPs of the YRB and the YEB are more significant than 1, indicating that the overall trend of ACE efficiencies is good, and the TFP of the YEB is slightly higher. The TFP of the YEB shows positive growth, the contribution of TP is positive, and the contribution of TE is negative. The TFP of the YRB is positive, the contribution of TP is positive, and the TE is almost unchanged. The contributions of TP in the YEB and YRB are more significant than those of TE, which indicates that the current ACE efficiencies in the two

**Table 9. Malmquist index table of the YEB and YRB.**

| Year | TFP in the YRB | TE in the YRB | TP in the YRB | TFP in the YEB | TE in the YEB | TP in the YEB |
|---|---|---|---|---|---|---|
| 2001–2002 | 1.0303 | 0.9676 | 1.0653 | 1.0256 | 0.9560 | 1.0753 |
| 2002–2003 | 1.0366 | 0.9653 | 1.0852 | 1.0023 | 0.9245 | 1.1038 |
| 2003–2004 | 1.1881 | 1.0613 | 1.1406 | 1.1690 | 0.9921 | 1.1794 |
| 2004–2005 | 1.0417 | 1.0800 | 0.9916 | 1.0112 | 0.9727 | 1.0401 |
| 2005–2006 | 1.0065 | 0.9765 | 1.0325 | 0.9849 | 0.9920 | 0.9931 |
| 2006–2007 | 1.2049 | 0.8588 | 1.4200 | 1.1788 | 0.8909 | 1.3433 |
| 2007–2008 | 1.1513 | 1.0494 | 1.1135 | 1.1083 | 1.0639 | 1.0558 |
| 2008–2009 | 0.9589 | 1.1547 | 0.8384 | 1.0051 | 1.0610 | 0.9489 |
| 2009–2010 | 1.0841 | 0.9989 | 1.1366 | 1.1527 | 1.0374 | 1.1253 |
| 2010–2011 | 1.0764 | 0.9149 | 1.1832 | 1.1618 | 0.9905 | 1.1729 |
| 2011–2012 | 1.0578 | 0.9813 | 1.0783 | 1.0379 | 0.9927 | 1.0418 |
| 2012–2013 | 1.0727 | 1.0363 | 1.0360 | 1.0423 | 0.9892 | 1.0667 |
| 2013–2014 | 1.0338 | 0.9979 | 1.0363 | 1.0604 | 1.0033 | 1.0552 |
| 2014–2015 | 1.0065 | 0.9955 | 1.0713 | 1.0692 | 0.9690 | 1.1025 |
| 2015–2016 | 1.0276 | 0.9393 | 1.1040 | 1.1779 | 1.0151 | 1.1608 |
| 2016–2017 | 1.0479 | 1.0239 | 1.0388 | 0.9892 | 0.9557 | 1.0373 |
| 2017–2018 | 1.0940 | 1.0320 | 1.0643 | 1.0274 | 0.9914 | 1.0363 |
| 2018–2019 | 1.1893 | 1.0063 | 1.1832 | 1.1776 | 1.0117 | 1.1830 |
| 2019–2020 | 1.6824 | 0.9679 | 1.7366 | 1.6249 | 1.1053 | 1.4998 |
| Mean value | 1.1048 | 1.0004 | 1.1240 | 1.1056 | 0.9955 | 1.1169 |

basins are technology-driven. TP can reduce the consumption of agricultural resources and reduce ACE, thus promoting the improvement of ACE efficiencies. However, with the continuous advancement of agricultural reform, it is more and more challenging to promote efficiency through reform, which causes a series of problems in agricultural environmental management and systems, restricting agricultural technical efficiency. Therefore, it is necessary to increase scientific research investment and promote scientific and technological innovation to promote the improvement of ACE efficiencies. On the other hand, effective measures should be taken to improve agricultural technical efficiency. Stages analyze the TFPs of the YEB and the YRB; the TFPs of the YEB and the YRB can be roughly divided into three stages. The index of the YEB fluctuated wildly from 2001 to 2011, the index change trend was stable from 2012 to 2018, and the index change showed a rapid growth trend from 2019 to 2020. Over the entire period, the TFP of the YEB was less than 1 in 2005–2006 and 2016–2017. The index of the YRB fluctuated wildly from 2001 to 2010, the index change trend was stable from 2011 to 2017, and the index change showed a rapid growth trend from 2018 to 2020. Over the entire period, the TFP of the YRB was less than 1 in 2008–2009.

## 5.3 Double dimensional analysis of total volume and efficiency of ACE

In order to further explore the relationship between total volume and efficiency of ACE, this part puts the two parts into the same coordinate system, takes the mean value of total volumes and efficiencies of 19 regions from 2001 to 2020 as the origin point, and divides 19 regions into four categories: low efficiency and low emission, low efficiency and high emission, high efficiency, and low emission, and the results are shown in Fig 6.

As shown in Fig 6, from the comparative analysis of the YEB and YRB, the YEB contains the largest proportion of the "high-emission and low-efficiency" zone. In contrast, the YRB contains the most significant proportion of the "low-emission and low-efficiency" area. Sichuan and Jiangsu, located in the YEB, belong to the "high-efficiency and high-emission" zone (the first quadrant). As the two provinces are important agricultural-producing areas, there are more carbon emissions in their production activities. However, the two provinces pay more attention to the rational allocation of input factors and have more desirable output than other regions. Therefore, the ACE efficiencies of the two provinces are also high when the ACE is high. Anhui, Jiangxi, Hubei, and Hunan in the YEB and Henan and Shandong in the YRB are located in the "high-emission and low-efficiency" zone (the second quadrant). Although these regions are also large agricultural provinces with significant carbon emissions, they have low ACE efficiency due to the emergence of problems such as no reasonable allocation of input factors or unreasonable industrial structure. These regions are crucial agricultural regulation areas and can learn from the agricultural development strategies of Sichuan and Jiangsu to help these areas get rid of "high-emission and low-efficiency " zones as soon as possible. Chongqing and Yunnan in the YEB and Gansu, Shanxi, Ningxia, and Inner Mongolia in the YRB are located in the "low-emission and low-efficiency" zone. Although the ACE in these regions is small, there is still redundant ACE, resulting in low ACE efficiency. This can be improved through reasonable waste treatment and optimization of agricultural industrial structures. At the same time, it can also refer to the agricultural policies and strategies in the "high-efficiency and low-emission" zone (the third quadrant) and choose the strategies suitable for developing their regions for continuous improvement. Shanghai, Zhejiang, and Guizhou in the YEB and Shaanxi and Qinghai in the YRB belong to the "low-emission and high-efficiency" zone (the fourth quadrant), which combines low emissions with high efficiency in agricultural production activities.

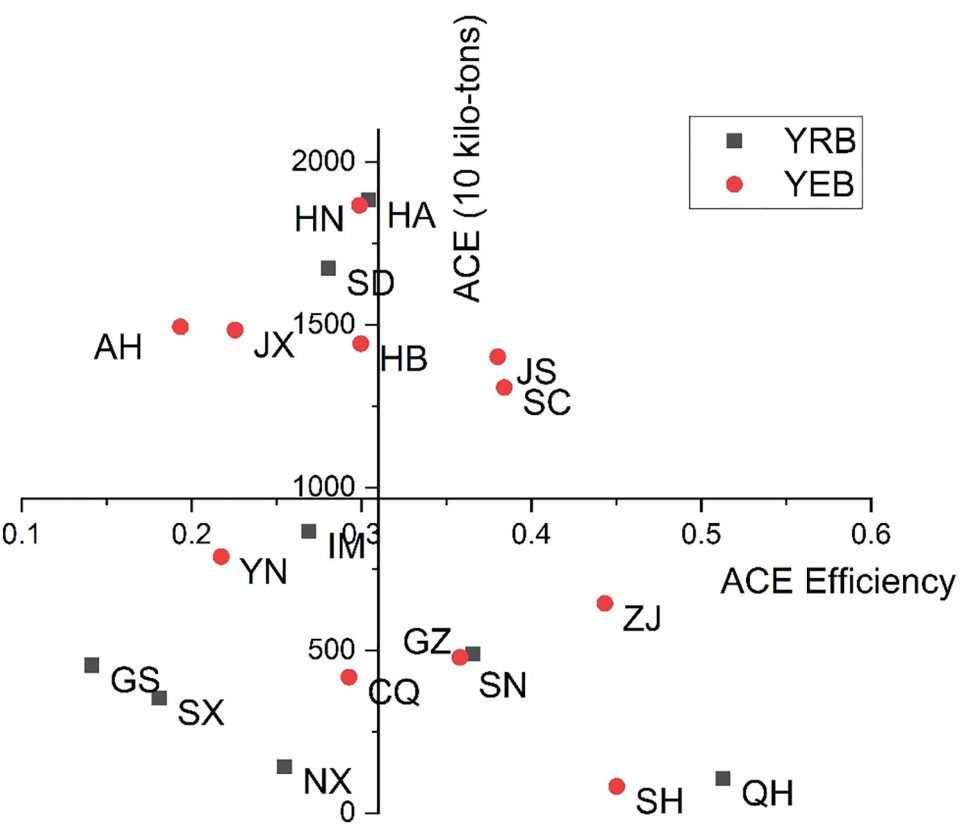

**Fig 6. Double dimensional map of ACE total volume and efficiency in different regions.**

## 5.4 CCD between ACE efficiency and green finance

**5.4.1 Measurement of green finance development level.** According to the index system established in Section 4.2, the entropy weight method in Section 3.4 is adopted to obtain the index weight, and the development levels of green finance in the YEB and the YRB are shown in Table 10.

Table 10 shows that, on average, the development level of green finance in the YEB is slightly more substantial than that in the YRB, and the level of green finance in the YEB is 0.1674, and that in the YRB is 0.1461. In terms of time, the level of green finance in the YEB decreases year by year from 2007 to 2009, shows a fluctuation increase from 2010 to 2013, reaches its highest point in 2013, and shows a trend of decline year by year from 2014 to 2020, reaching 0.1449 in 2020. The level of green finance in the YRB shows an increasing trend from 2007 to 2008, a decreasing trend year by year from 2009 to 2011, and a fluctuating trend after 2012. After 2007, the state introduced a series of policies to promote the development of green finance. However, due to the financial shock 2008, the level of green finance in 2009 decreased significantly, and the impact on the YRB is more significant than that on the YEB. Generally speaking, the development level of green finance from 2007 to 2020 fluctuates, which is approximately "inverted U-shaped." The fluctuation of the development level of green finance indicates that the development of green finance needs to be more mature, the institutional construction of green finance development needs to be better, and the indicators of green finance are greatly affected by market and policy changes. There is still room for progress in the future.

**Table 10. The levels of green finance in the YEB and the YRB.**

| Year | The levels of green finance in the YEB | The levels of green finance in the YRB |
|---|---|---|
| 2007 | 0.159461 | 0.166689 |
| 2008 | 0.151172 | 0.176205 |
| 2009 | 0.14394 | 0.138291 |
| 2010 | 0.173762 | 0.115991 |
| 2011 | 0.175803 | 0.113635 |
| 2012 | 0.159809 | 0.127763 |
| 2013 | 0.239424 | 0.170508 |
| 2014 | 0.194719 | 0.156666 |
| 2015 | 0.187878 | 0.133079 |
| 2016 | 0.161255 | 0.151094 |
| 2017 | 0.155972 | 0.143825 |
| 2018 | 0.151948 | 0.141909 |
| 2019 | 0.143328 | 0.162822 |
| 2020 | 0.144925 | 0.147035 |
| Mean value | 0.167386 | 0.146108 |

**5.4.2 Analysis of CCD between ACE efficiency and green finance level.** The CCD of ACE efficiency and green finance development level is calculated by combining the formula in Section 3.5, and the grades of CCD are evaluated, as shown in Table 11. For layout reasons, only the results of the comprehensive evaluation for 2007 and 2020 are presented.

**Table 11. The CCD of ACE efficiency and green finance level in different regions.**

| Regions | 2007 | | 2020 | | Rate of change (%) |
|---|---|---|---|---|---|
| | CCD | Rrade | CCD | Rrade | |
| QH | 0.5111 | barely coordination | 0.6794 | primary coordination | 32.93 |
| NX | 0.3601 | mild disorder | 0.4641 | near disorder | 28.88 |
| GS | 0.3960 | mild disorder | 0.5792 | barely coordination | 46.26 |
| IM | 0.4162 | near disorder | 0.5742 | barely coordination | 37.96 |
| HA | 0.4666 | near disorder | 0.5976 | barely coordination | 28.08 |
| SN | 0.5011 | barely coordination | 0.6402 | primary coordination | 27.76 |
| SX | 0.3954 | mild disorder | 0.4885 | near disorder | 23.55 |
| SD | 0.3657 | mild disorder | 0.4741 | near disorder | 29.64 |
| YRB | 0.4265 | near disorder | 0.5622 | barely coordination | 31.82 |
| SC | 0.5045 | barely coordination | 0.5569 | barely coordinated | 10.39 |
| CQ | 0.4877 | near disorder | 0.6235 | primary coordination | 27.84 |
| GZ | 0.3394 | mild disorder | 0.5409 | barely coordination | 59.37 |
| YN | 0.3282 | mild disorder | 0.4302 | near disorder | 31.08 |
| HB | 0.4555 | near disorder | 0.6120 | primary coordination | 34.36 |
| HN | 0.3783 | mild disorder | 0.6206 | primary coordination | 64.05 |
| AH | 0.3297 | mild disorder | 0.4994 | near disorder | 51.47 |
| JX | 0.4544 | near disorder | 0.5040 | barely coordination | 10.92 |
| JS | 0.4439 | near disorder | 0.6463 | primary coordination | 45.60 |
| ZJ | 0.4702 | near disorder | 0.6197 | primary coordination | 31.79 |
| SH | 0.4152 | near disorder | 0.6097 | primary coordination | 46.84 |
| YEB | 0.4188 | near disorder | 0.5694 | barely coordination | 35.96 |
| Mean value | 0.4221 | near disorder | 0.5663 | barely coordination | 34.18 |

As can be seen from Table 9, in 2007, all provinces were in the stage of mild disorder, near disorder, and barely coordination, and 2020, they reached the stage of near disorder, barely coordination, and primary coordination. In 2007, the mean value of CCDs in 19 provinces was near disorder, and in 2020, it reached bare coordination. Both the YEB and the YRB have jumped from near disorder in 2007 to barely coordinated status in 2020. A benign coupling relationship exists between ACE efficiency and green finance development level in the YEB and the YRB. In 2007, the CCD of the YEB was slightly lower than that of the YRB, but it was slightly higher in 2020, indicating that the CCD of the YEB rises slightly faster. There are regional differences in the CCD between the YEB and the YRB. In the YRB, Qinghai has the highest CCD, reaching 0.6794 in 2020. Ningxia has the lowest CCD, reaching 0.4641 in 2020. Qinghai and Shaanxi reached the primary coordination level in 2020; the rest were near disorder and barely coordination level. The CCD of the YEB is the highest in Jiangsu, reaching 0.6463 in 2020 and the lowest in Yunnan, reaching 0.4302 in 2020. In the YEB, Chongqing, Hubei, Hunan, Jiangsu, Zhejiang, and Shanghai reached the primary coordination level in 2020, and the rest were near disorder and barely coordination level.

In order to better observe whether there are spatial differences of the CCDs in these regions, the research period is divided into two stages (2007–2013 and 2014–2020), and the mean values of the CCD in the two stages are calculated respectively to determine the coupling coordination grades. The results are shown in Table 12.

It can be observed from Table 12 that there are spatial differences in CCD between the two periods, that is, CCD of downstream > CCD of midstream > CCD of upstream. From 2007 to 2013, 40% of the downstream provinces were in the bare coordination grade, 40% near disorder grade, and 20% in mild disorder grade. In the midstream, 33.33% of the provinces were in the barely coordination grade, 50% were in the near disorder grade, and 16.7% were in the mild disorder grade. In the upstream, 25% of all provinces were in the barely coordination grade, 25% were in the near disorder grade, and 50% were in the mild disorder grade. From 2014 to 2020, 20% of the downstream provinces were at the primary coordination grade, 60% were at the barely coordination grade, and 20% were in the near disorder grade. In the midstream, 33.33% of the provinces were in the barely coordination grade, and 66.67% were near disorder grade. In the upstream, 37.5% of all provinces were in the barely coordination grade, 50% were in the near disorder grade, and 12.5% were in the mild disorder grade. Compared with the period from 2007 to 2013, the CCD of each province has improved to a certain extent from 2014 to 2020. The number of provinces with primary coordination has changed from 0 to 1, the number of provinces with barely coordination has increased by 2, the number of provinces with near disorder has increased by 2, and the number of provinces with mild disorder has decreased by 5. The CCD of the YEB is higher than that of the YRB, and the CCD of the Yangtze River delta is the highest in the YEB.

**Table 12. Evolution of spatial pattern of CCD in different regions.**

| Year | Coordination level | Upstream | Midstream | Downstream |
|------|-------------------|----------|-----------|------------|
| 2007–2013 | Mild disorder | NX、GS、GZ、YN | AH | SD |
| | Near disorder | QH、IM | SX、HN、JX | HA、JS |
| | barely coordination | SC、CQ | SN、HB | ZJ、SH |
| 2014–2020 | Mild disorder | YN | | |
| | Near disorder | NX、GS、IM、GZ | SX、HN、AH、JX | SD |
| | barely coordination | QH、SC、CQ | SN、HB | HN、JS、ZJ |
| | primary coordination | | | SH |

## 5.5 CCD between ACE efficiency and green finance

In this part, the GRA method in Section 3.6 is adopted to measure the degree of effect of each influencing factor on the CCD. The mean values of the regions included in the YRB and the region included in the YEB are taken as the degree of effect of each influencing factor in the YRB and the YEB on the CCD. The calculation results are visualized with the origin software, as shown in Fig 7. The greater the grey correlation degree, the greater the effect of this factor on the CCD.

From an overall perspective, the effects of the influencing factors on the CCD of the two systems from large to small are as follows: human capital quality, economic development level, government regulation, and scientific and technological innovation ability. Among them, the human capital qualities of the YEB and YRB have the most significant influence on the CCD of the two systems, and from 2007 to 2020, the influence degree is relatively stable, maintaining around 0.8. From 2007 to 2020, the influence degrees of the other three factors on the CCD show trends of "Ω-shaped," which reached peaks around 2014 and then declined and increased to a certain extent in 2019–2020. It can be seen that when the CCD is at the disorder grade, improving the quality of human capital, the level of economic development, the ability of scientific and technological innovation, and stimulating the macro-control of the government can effectively improve the CCD of the two systems. However, when the CCD of the two systems enters the coordination grade, improving the level of economic development, scientific and technological innovation ability, and stimulating the macro-control of the government cannot effectively improve the CCD, but only improving the quality of human capital can effectively improve the CCD of the two systems.

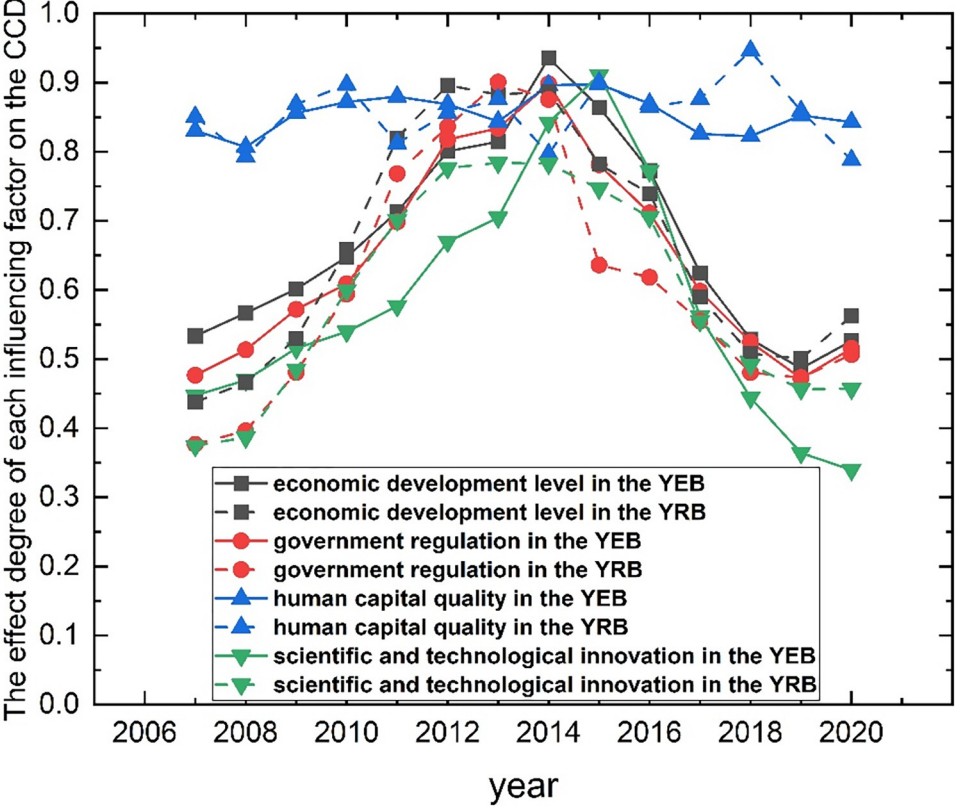

**Fig 7. The effect degree of each influencing factor on the CCD in the YEB and YRB.**

From the comparison of the YEB and YRB, the degree of influence of each factor is different to some extent. The effect of the same factors on the CCD of the YEB and the YRB differs.

The effects of human capital quality on the CCD of the YEB and the YRB fluctuate around 0.7–0.9, but the fluctuation amplitudes differ. Among them, there were greater degrees of impact on the YRB in 2010, 2013, 2017, and 2018. Meanwhile, there are greater degrees of impact on the YEB in 2011,2014, and 2020 and similar degrees of impact on the YEB and YRB in the remaining years. From the average YEB and YRB values, human capital quality had little effect on the CCD in the YEB and the YRB from 2007 to 2020. However, the effect of human capital quality on the YEB is more stable, and the fluctuation is more minor.

Although the influence of economic development levels on the CCD of the YEB and the YRB is "Ω-shaped," there are differences in the trends. The YEB had an upward trend from 2007 to 2014, reached a peak in 2014, and then a downward trend until an upward trend in 2020. However, in the YRB, the trend increased from 2007 to 2012, stabilized for two years, decreased from 2014 to 2019, and increased in 2020. The economic development levels in 2010–2013 and 2019–2020 had slightly higher effects on the YRB, and the other years had slightly higher effects on the YEB. From the average point of view, the economic development level from 2007 to 2020 had a slightly higher impact on the CCD of the two systems in the YEB. The main reason is that compared with the YRB, the economic development level of the YEB is higher, which can provide a solid material foundation for reducing ACE and developing green finance to promote the coupling and coordinated development of the two systems.

The influence degree trends of government regulation on the CCD of the YEB and the YRB are different. The YEB had an upward trend from 2007 to 2014, a downward trend from 2014 to 2019, and an upward trend in 2020, while the YRB had an upward trend from 2007 to 2013, a downward trend from 2013 to 2019, and an upward trend in 2020. From 2011 to 2014, the effect of government regulation on the YRB was higher than that of the YEB, and in other years, it was slightly higher in the YEB. Compared with the YRB, the effect of government regulation on the CCD in the YEB from 2007 to 2020 was slightly higher. As the economy of the YEB is relatively developed, the financial revenue of the regional government is relatively sufficient. Hence, the financial expenditure is significant, and the government has more expenditure to improve the CCD of the two systems.

There are differences in the effect degree trend of scientific and technological innovation ability on the YEB and YRB CCD. The YEB had an upward trend from 2007 to 2015 and a downward trend from 2015 to 2020, while the YRB had an upward trend from 2007 to 2012, a stable trend from 2012 to 2014, and a downward trend from 2014 to 2020. In 2010–2013 and 2018–2020, the effect of scientific and technological innovation ability on the YRB was higher than that of the YEB, and YEB was slightly higher in other years. From the average point of view, the coupling degree effect of government regulation on the YRB is slightly higher than that of the YEB during 2007–2020. Compared with the YEB, the YRB develops more slowly, and its scientific and technological level needs to be revised. Improving scientific and technological innovation ability can promote the coupling and coordinated development of the two systems.

## 6. Conclusions

Through the above empirical research, this paper draws the following conclusions:

1. From the perspective of ACE and ACI, the YEB and the YRB show a trend of "rising first and then falling." The total ACE of the YEB is roughly twice that of the YRB, and there is still much room for emission reduction, but the YEB reaches its peak two years earlier than the YRB. From the perspective of growth rate, the YEB and YRB show trends of fluctuating

decline, and the decline rate of the YEB is faster, indicating that China's agriculture is in the transition stage from high carbon to low carbon. From the perspective of ACI, the ACIs of the YEB and YRB show trends of continuous decline; the decline rate of the YRB is faster, and the agricultural productions and lifestyles of the YEB and YRB are constantly developing in good trends.

2. Analysis of ACE efficiency from a static perspective: During the study period, the ACE efficiencies of the YEB and YRB show trends of overall increase, and the YEB is slightly higher. According to the analysis of the regions within the YEB and YRB, except for the fluctuation of Shanghai and Qinghai, the ACE efficiencies in other regions show continuous improvement trends, and the increase rates are significantly different in different regions. From the dynamic point of view, the TFPs of the YEB and YRB are more significant than 1, and the YEB is slightly higher. The contributions of TP in the YEB and YRB are more significant than those of TE, and TP has a greater impact on TFP.

3. From the dual dimensions of total volume and efficiency, it can be found that the YEB contains the most significant proportion of the "high-emission and low-efficiency" zone, and the YRB contains the most significant proportion of the "low-emission and low-efficiency" zone. In the YEB, Jiangsu and Sichuan are in the "high-emission and high-efficiency" zone; Anhui, Jiangxi, Hubei, and Hunan are in the "high-emission and low-efficiency" zone; Chongqing and Yunnan are in the "low-emission and low-efficiency" zone; Shanghai, Zhejiang and Guizhou are in the "low-emission and high-efficiency" zone. In the YRB, Henan and Shandong are in the "high-emission and low-efficiency" zone; Gansu, Shanxi, Ningxia, and Inner Mongolia are in the "low-emission and low-efficiency" zone; Shaanxi and Qinghai are in the "low-emission and high-efficiency" zone.

4. From the perspective of the CCD of ACE efficiency and green finance level, the current ACE efficiencies of the YEB and the YRB have relatively benign coupling relationships with the development level of green finance. The CCD of the YEB has increased slightly faster, and there are regional differences in the CCDs of the YEB and YRB. The CCDS of the YEB and YRB are the highest in Jiangsu and Qinghai. The CCDs of the two time periods from 2007 to 2013 and 2014 to 2020 have spatial differences: downstream CCD > midstream CCD > upstream CCD.

5. From the perspective of influencing factors of the CCD of the two systems, the influence ranges from large to small: Human capital quality, economic development level, government regulation, scientific and technological innovation ability, from the comparison of the YEB and YRB, only scientific and technological innovation ability has a more significant impact on the YRB than on the YEB, the economic development level and government regulation have more significant impacts on the CCD of the two systems in the YEB, and the human capital quality has a similar effect on the CCD of the YEB and YRB.

Although the research has obtained considerable results, there are still some shortcomings.

1. From a broad perspective, agriculture represents agriculture, forestry, animal husbandry, and fishery. When calculating agricultural carbon emissions, this paper only considers planting and animal husbandry but fails to consider forestry and fishery. Therefore, there is a specific deviation between the measured ACE and the actual ACE. In the future, the index system of forestry and fishery carbon emissions can be clarified, and the carbon emissions of forestry and fishery can be quantified so that the measured ACE can be as close as possible to the actual carbon emissions.

2. This paper only analyzes the ACE efficiencies of the YEB and the YRB from the provincial perspective, which is difficult to analyze from the city perspective due to the difficulties in obtaining data. In the future, the data needed for ACE in each city can be obtained through different data acquisition channels and field investigations, and then the ACE in the YEB and the YRB can be analyzed from the perspective of the city so that more detailed, specific and compelling measures can be proposed for the ACE in each city to promote the reduction of ACE and improve ACE efficiency.

## 7. Policy implications

1. To adapt to local conditions, develop agriculture with characteristics, and adjust the agricultural industrial structure. Due to differences in geographical conditions and national positioning, the total ACEs of the YEB and the YRB are different. The regions within the YEB and the YRB should promote the reform of agricultural industrial structure according to local conditions and reduce agricultural emissions. For regions with similar geographical and climatic conditions, the regions with lower efficiency should learn from the experience of the regions with higher efficiency to promote ACE reduction.

2. To increase investment in agricultural research and improve scientific and technological innovation ability to improve ACE efficiency. TP is the main factor of TFP growth of ACE in the YEB and YRB. The government should increase investment in agricultural scientific research, develop fertilizers, pesticides, and other agricultural materials with low carbon emissions, improve production facilities and other measures, and give full play to scientific and technological innovation in promoting TFP.

3. To accelerate the construction of the green financial system, optimize the application of green financial product portfolio, innovate green financing models, and vigorously support the transformation and development of agriculture. Finance should increase investment in environmentally friendly rural agricultural activities, supplement the funding gap for the green development of rural agriculture to help farmers carry out agricultural transformation faster, and accelerate the emission reduction and carbon fixation in agriculture.

4. To improve the quality of rural human capital. On the one hand, more talents can be attracted to stay in rural areas, and the quality of human capital in rural areas can be improved through policy inclination and assistance policies. The application of advancing research results, policies, and concepts by agricultural talents can not only promote the development of green finance by enabling more environmentally friendly projects to be developed through the financing of green finance but also improve the efficiency and quality of agricultural production, improve the efficiency of carbon emission, and promote the coupling and coordinated development of the two systems. On the other hand, training high-quality farmers, a precise selection of cultivation objects, strengthening the construction of the cultivation system and promoting the quality and efficiency of the cultivation work.

## Supporting information

**S1 File.**
(DOCX)

## Acknowledgments

The authors would like to sincerely thank the editor and reviewers for their kind comments.

## Author Contributions

**Conceptualization:** Jingjie Li, Chenying Cui.

**Data curation:** Chenying Cui.

**Formal analysis:** Jingjie Li, Chenying Cui.

**Funding acquisition:** Jingjie Li.

**Investigation:** Chenying Cui.

**Methodology:** Jingjie Li, Chenying Cui.

**Project administration:** Jingjie Li.

**Resources:** Chenying Cui.

**Software:** Chenying Cui.

**Supervision:** Jingjie Li.

**Validation:** Jingjie Li.

**Visualization:** Chenying Cui.

**Writing – original draft:** Jingjie Li, Chenying Cui.

**Writing – review & editing:** Jingjie Li.

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
