## [Decision Letter · Decision Letter 0]

24 Apr 2024

PONE-D-24-07132Analysis of spatiotemporal changes, influencing factors, and coupling coordination of agricultural carbon emission efficiency in river basins of ChinaPLOS ONE

Dear Dr. Li,

Thank you for submitting your manuscript to PLOS ONE. After careful consideration, we feel that it has merit but does not fully meet PLOS ONE’s publication criteria as it currently stands. Therefore, we invite you to submit a revised version of the manuscript that addresses the points raised during the review process.

 Please submit your revised manuscript by Jun 08 2024 11:59PM. If you will need more time than this to complete your revisions, please reply to this message or contact the journal office at plosone@plos.org. Please include the following items when submitting your revised manuscript:A rebuttal letter that responds to each point raised by the academic editor and reviewer(s). You should upload this letter as a separate file labeled 'Response to Reviewers'.A marked-up copy of your manuscript that highlights changes made to the original version. You should upload this as a separate file labeled 'Revised Manuscript with Track Changes'.An unmarked version of your revised paper without tracked changes. You should upload this as a separate file labeled 'Manuscript'.

We look forward to receiving your revised manuscript.

Kind regards,

Hesham M.H. Zakaly, Ph.D.

Academic Editor

PLOS ONE

Journal Requirements:

   "This work is supported by The Tianjin Philosophy and Social Science Planning

Program, China (Grant numbers TJTJ23-001)"

5. We note that your Data Availability Statement is currently as follows: All relevant data are within the manuscript and its Supporting Information files.

6. We note that Figure 6 and 7 in your submission contain map/satellite images which may be copyrighted. All PLOS content is published under the Creative Commons Attribution License (CC BY 4.0), which means that the manuscript, images, and Supporting Information files will be freely available online, and any third party is permitted to access, download, copy, distribute, and use these materials in any way, even commercially, with proper attribution. For these reasons, we cannot publish previously copyrighted maps or satellite images created using proprietary data, such as Google software (Google Maps, Street View, and Earth). For more information, see our copyright guidelines: http://journals.plos.org/plosone/s/licenses-and-copyright.

a. You may seek permission from the original copyright holder of Figure 6 and 7 to publish the content specifically under the CC BY 4.0 license.  

Reviewers' comments:

Reviewer's Responses to Questions

**Comments to the Author**

1. Is the manuscript technically sound, and do the data support the conclusions?

Reviewer #1: Yes

Reviewer #2: Yes

2. Has the statistical analysis been performed appropriately and rigorously? 

Reviewer #1: Yes

Reviewer #2: Yes

3. Have the authors made all data underlying the findings in their manuscript fully available?

Reviewer #1: Yes

Reviewer #2: Yes

4. Is the manuscript presented in an intelligible fashion and written in standard English?

Reviewer #1: Yes

Reviewer #2: Yes

5. Review Comments to the Author

Reviewer #1: 1. Please add line numbers in the manuscript. The current version does not have line numbers and cannot clearly describe the specific areas that need to be modified in the article.

2. Is there no citation of references in the introduction section? Please add relevant literature references.

3. “used the IPCC carbon emission coefficient method to measure ACE in Shandong Province from 2000 to 2020”, the u in “used” needs to be capitalized.

4. It is strongly recommended to organize the abbreviations of nouns in the text into a table at the beginning of the manuscript.

5. “with Heze City ranking first..”. An additional period has been added.

6. The literature review only briefly lists the research and does not provide its own summary and viewpoint, which is not a qualified literature review. Please revise and summarize carefully

7. Before 2015, agricultural production, especially planting activities, in the Yangtze and Yellow River basins were still mainly focused on small-scale farming, while intensive agricultural production was mainly concentrated in the northeast and northwest China. However, the majority of young and middle-aged people in rural areas enter cities in search of higher incomes and rarely directly participate in agricultural activities. So why choose indicator 3 for human resources and indicator 4 for green production efficiency in enterprises?

8. The results and discussion suggestions are separated, and the current fifth chapter is too long.

9. Please carefully check the format and content expression of the article. Currently, some expressions are too colloquial and there are issues with inconsistent formatting.

10. Please modify the image color scheme and table format, as the image title is too short.

Reviewer #2: The study have valuable insights into the spatiotemporal dynamics, influencing factors, and coupling coordination of agricultural carbon emission efficiency in the YEB and YRB regions in China. It highlights the importance of technological progress, human capital quality, economic development level, government regulation, and scientific and technological innovation ability in influencing the coordination between ACE efficiency and green finance. These findings have implications, aiming to promote sustainable agricultural systems and achieve carbon neutrality in China's river basins.

However, there are some comments and inquiries as follows:

Major comments

1-What is the novelty in your article?

2-How is your article different from the following articles:

a-Coupling Coordination and Spatiotemporal Dynamic Evolution between Agricultural Carbon Emissions and Agricultural Modernization in China 2010–2020

b-The Coupling and Coordination of Agricultural Carbon Emissions Efficiency and Economic Growth in the Yellow River Basin, China

c-Conflict or Coordination? Analysis of Spatio-Temporal Coupling Relationship between Urbanization and Eco-Efficiency: A Case Study of Urban Agglomerations in the Yellow River Basin, China

3-The title needs to be upgraded. What about this title “Spatiotemporal Trends and Coordination of Agricultural Carbon Efficiency in YEB and YRB River Basins, China: An Analysis of Influencing Factors and Green Finance Integration”

What about the investigations after 2020?

4-What about the depth of coupling coordination?

I mean the study evaluates CCD between ACE efficiency and green finance but not investigates deeply the specific mechanisms, policies, or strategies that can enhance coordination or improve agricultural carbon emission efficiency.

Minor comments

In Figure 2, the X-axis title is in Chinese language.

In Table 3, Capitalize the items in “Carbon emission resources”.

In Table 6, add the title of year column.

6. PLOS authors have the option to publish the peer review history of their article (what does this mean?). If published, this will include your full peer review and any attached files.

Reviewer #1: No

Reviewer #2: No

---

## [Author Response · Author response to Decision Letter 0]

2 Jun 2024

Dear editor and reviewers, 

Thank you for your careful review and constructive suggestions regarding our manuscript. We have revised the manuscript in accordance with the comments and marked all the amends on our revised manuscript. Please refer to the attached "Response to Reviewers" and "Manuscript" for specific modifications.

Best wishes,

Jingjie Li

---

## [Decision Letter · Decision Letter 1]

25 Jun 2024

PONE-D-24-07132R1Spatiotemporal Trends and Coordination of Agricultural Carbon Efficiency in the Yangtze River Economic Belt and Yellow River Basin, China: An Analysis of Influencing Factors and Green Finance IntegrationPLOS ONE

Dear Dr. Li,

Thank you for submitting your manuscript to PLOS ONE. After careful consideration, we feel that it has merit but does not fully meet PLOS ONE’s publication criteria as it currently stands. Therefore, we invite you to submit a revised version of the manuscript that addresses the points raised during the review process.

We look forward to receiving your revised manuscript.

Kind regards,

Hesham M.H. Zakaly, Ph.D.

Academic Editor

PLOS ONE

Reviewers' comments:

Reviewer's Responses to Questions

**Comments to the Author**

1. If the authors have adequately addressed your comments raised in a previous round of review and you feel that this manuscript is now acceptable for publication, you may indicate that here to bypass the “Comments to the Author” section, enter your conflict of interest statement in the “Confidential to Editor” section, and submit your "Accept" recommendation.

Reviewer #2: (No Response)

2. Is the manuscript technically sound, and do the data support the conclusions?

Reviewer #2: (No Response)

3. Has the statistical analysis been performed appropriately and rigorously? 

Reviewer #2: (No Response)

4. Have the authors made all data underlying the findings in their manuscript fully available?

Reviewer #2: (No Response)

5. Is the manuscript presented in an intelligible fashion and written in standard English?

Reviewer #2: (No Response)

6. Review Comments to the Author

Reviewer #2: The author did not find any replies to his comments in the attached manuscript as the authors stated. The reviewer insists on the writing of each of his comment attached with the response of author.

7. PLOS authors have the option to publish the peer review history of their article (what does this mean?). If published, this will include your full peer review and any attached files.

Reviewer #2: No

---

## [Author Response · Author response to Decision Letter 1]

1 Jul 2024

Dear editor and reviewers, 

Thank you for your review. We have revised the manuscript in accordance with the comments and marked all the amends on our revised manuscript. The following is a list of responses to each suggestion. 

Reviewer #2: The author did not find any replies to his comments in the attached manuscript as the authors stated. The reviewer insists on the writing of each of his comment attached with the response of author.

Response: Dear Reviewer 2, in the first revision version, according to the requirements of Plos One, we have submitted three revised documents: (1) A revised letter that responds to each point raised by the academic editor and reviewer (s) labeled 'Cover letter. (2) A marked-up copy of the manuscript highlighting changes made to the original version, labeled 'Revised Manuscript with Track Changes'. (3) An unmarked version of the revised paper without tracked changes, labeled 'Manuscript'.

Previously, we placed Response to Reviewers in the "cover letter" folder. This time, we have placed Response to Reviewers in a dedicated "Response to Reviewers" folder. It is at the end of the system generated file "PONE-D-24-07132-R2".

In addition, in order to have a clearer understanding of the modifications for you, we have acceptance of amendments made to Reviewer 1 (another reviewer) in the Revised Manuscript with Track Changes document. Therefore, this version of the Revised Manuscript with Track Changes is only based on your previous comment.

We apologize for any inconvenience caused and thank you again for your review. We deeply appreciate your consideration of our manuscript, and we look forward to receiving comments.

Thank you very much.

Best Regards.

Yours Sincerely,

Jingjie Li

Associate Professor of School of Science, Tianjin University of Commerce, China

Executive Director of Tianjin Mathematics Association

Member of Tianjin Industrial and Applied Mathematics Society

Supervisor of Master. students, School of Science, Tianjin University of Commerce.

---

## [Decision Letter · Decision Letter 2]

24 Jul 2024

Spatiotemporal Trends and Coordination of Agricultural Carbon Efficiency in the Yangtze River Economic Belt and Yellow River Basin, China: An Analysis of Influencing Factors and Green Finance Integration

PONE-D-24-07132R2

Dear Dr. Li,

We’re pleased to inform you that your manuscript has been judged scientifically suitable for publication and will be formally accepted for publication once it meets all outstanding technical requirements.

Kind regards,

Hesham M.H. Zakaly, Ph.D.

Academic Editor

PLOS ONE

Additional Editor Comments (optional):

Reviewers' comments:

Reviewer's Responses to Questions

**Comments to the Author**

1. If the authors have adequately addressed your comments raised in a previous round of review and you feel that this manuscript is now acceptable for publication, you may indicate that here to bypass the “Comments to the Author” section, enter your conflict of interest statement in the “Confidential to Editor” section, and submit your "Accept" recommendation.

Reviewer #2: All comments have been addressed

2. Is the manuscript technically sound, and do the data support the conclusions?

Reviewer #2: Partly

3. Has the statistical analysis been performed appropriately and rigorously? 

Reviewer #2: Yes

4. Have the authors made all data underlying the findings in their manuscript fully available?

Reviewer #2: Yes

5. Is the manuscript presented in an intelligible fashion and written in standard English?

Reviewer #2: Yes

6. Review Comments to the Author

Reviewer #2: (No Response)

7. PLOS authors have the option to publish the peer review history of their article (what does this mean?). If published, this will include your full peer review and any attached files.

Reviewer #2: No

---

## [Editor Report · Acceptance letter]

20 Aug 2024

PONE-D-24-07132R2 

PLOS ONE

Dear Dr. Li, 

I'm pleased to inform you that your manuscript has been deemed suitable for publication in PLOS ONE. Congratulations! Your manuscript is now being handed over to our production team.

Kind regards, 

on behalf of

Dr. Hesham M.H. Zakaly 

Academic Editor

PLOS ONE